# LiftedCL: Lifting Contrastive Learning for Human-Centric Perception

**Ziwei Chen[1,2], Qiang Li[3]∗, Xiaofeng Wang[4], Wankou Yang[1,2]∗**

[1] School of Automation, Southeast University [2] Key Lab of Measurement and Control
of Complex Systems of Engineering, Ministry of Education, Southeast University, Nanjing, China
[3] Y-tech, Kuaishou Technology [4] Institute of Automation, Chinese Academy of Sciences
{richard_chen, wkyang}@seu.edu.cn  liqiang03@kuaishou.com
wangxiaofeng2020@ia.ac.cn

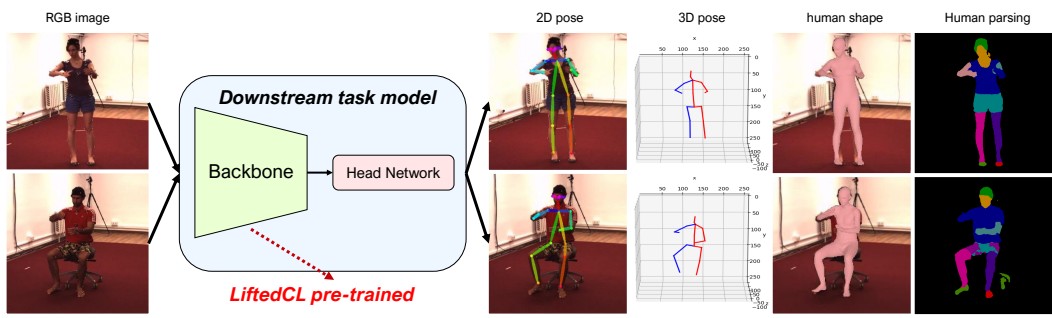

Figure 1: The backbone network pre-trained by LiftedCL can be transferred to various human-centric downstream tasks including human pose estimation, human shape recovery and human parsing. The first column shows two examples from the Human3.6M (Ionescu et al., 2013) dataset. The second and third columns present the estimated 2D and 3D pose. The last two columns demonstrate the reconstructed human mesh and the estimated human semantic parts.

## Abstract

Human-centric perception targets for understanding human body pose, shape and segmentation. Pre-training the model on large-scale datasets and fine-tuning it on specific tasks has become a well-established paradigm in human-centric perception. Recently, self-supervised learning methods have re-investigated contrastive learning to achieve superior performance on various downstream tasks. When handling human-centric perception, there still remains untapped potential since 3D human structure information is neglected during the task-agnostic pre-training. In this paper, we propose the Lifting Contrastive Learning (LiftedCL) to obtain 3D-aware human-centric representations which absorb 3D human structure information. In particular, to induce the learning process, a set of 3D skeletons is randomly sampled by resorting to 3D human kinematic prior. With this set of generic 3D samples, 3D human structure information can be learned into 3D-aware representations through adversarial learning. Empirical results demonstrate that LiftedCL outperforms state-of-the-art self-supervised methods on four human-centric downstream tasks, including 2D and 3D human pose estimation (0.4% mAP and 1.8 mm MPJPE improvement on COCO 2D pose estimation and Human3.6M 3D pose estimation), human shape recovery and human parsing.

## 1 Introduction

Human-centric perception, such as human pose estimation (Xiao et al., 2018; Sun et al., 2019; Pavllo et al., 2019; Gong et al., 2021), human shape recovery (Kanazawa et al., 2018; Choi et al., 2020; Xu et al., 2021) and human parsing (Yang et al., 2019; Li et al., 2020; Gong et al., 2018), has received significant attention in computer vision. Similar to other computer vision tasks, pre-training the

---

∗Corresponding authors.

model has become a widely-used paradigm in human-centric perception. Generally, models are first pre-trained on large-scale datasets (e.g., ImageNet (Deng et al., 2009)) and then fine-tuned on specific human-centric downstream task.

For human-centric perception, leveraging 3D human structure information on fine-tuning stage has been demonstrated effective to improve the performance. For instance, in the task of 3D pose estimation. RepNet (Wandt & Rosenhahn, 2019) adds a KCS (Wandt et al., 2018) layer into an adversary to better represent bone lengths and joint angles of a pose, which achieves more accurate 3D pose reconstruction results. HMR (Kanazawa et al., 2018) employs a prior human body model parameterized by shape and 3D joint angles and shows competitive results on 3d pose estimation and part segmentation. In (Qiu et al., 2019), a penalty is be added when the estimated 3D pose has unreasonable limb lengths according to the human body structure prior. Such methods show that leveraging 3D human kinematic prior on fine-tuning stage contributes to the performance. We argue that models can also benefit from 3D human kinematic prior on pre-training stage.

Concurrently, powered by contrastive representation learning, recent self-supervised pre-training methods (Chen et al., 2020a; He et al., 2020; Grill et al., 2020; Caron et al., 2018) have broken the dominance of supervised ImageNet pre-training on various downstream tasks, including image classification, object detection, semantic segmentation, etc. These self-supervised learning methods mainly adapt the image-level instance discrimination formulation as a pretext task to learn transferable representations. Besides, some methods (Wang et al., 2021a; Xie et al., 2021; Xiao et al., 2021; Wang et al., 2021b) extend the image-level contrastive learning framework to a dense paradigm, achieving superior performance on dense prediction tasks.

However, existing contrastive representation learning methods are mainly designed for image classification, object detection and semantic segmentation, rather than human-centric perception. Better results in these tasks may not guarantee superior performance in human-centric perception (see Table 4). Moreover, most of these works do not link 3D prior to 2D representation learning. In human-centric perception, there exist some challenging cases, for example, invisible joints, self-occluded keypoints, in which 3D human kinematic prior can be utilized to help better understand the relationship between body parts. Thus, it is still desirable to have a pre-training approach for human-centric tasks. Our goal is to improve contrastive learning by leveraging 3D human structure information for human-centric pre-training in a simple yet effective way.

To this end, we propose a novel contrastive learning framework termed LiftedCL to exploit 3D human structure information for human-centric pre-training. Firstly, we generalize the conventional InfoNCE loss (Oord et al., 2018) to an equivariant paradigm. Based on this, image-level invariant and pixel-level equivariant contrastive learning are applied to the projected feature vectors and maps respectively. Meanwhile, the representations are transformed into 3D human skeleton to better reveal the hidden human structure information. In particular, a set of 3D skeletons is randomly sampled by resorting to 3D human kinematic prior. With this set of real 3D samples, an adversary is adopted to induce the learning of 3D-aware human-centric representations.

We demonstrate the effectiveness of our proposed LiftedCL by pre-training using MS COCO (Lin et al., 2014) human images and fine-tuning on specific target dataset. Compared to the state-of-the-art method PixPro (Xie et al., 2021), LiftedCL achieves significant improvements on various human-centric downstream tasks, including COCO 2D human pose estimation (+0.4% mAP), MPII 2D human pose estimation (+0.3% PCKh@0.5), Human3.6M 2D human pose estimation (+0.9% JDR), Human3.6M 3D human pose estimation (1.8mm MPJPE), 3DPW human shape recovery (1.7mm reconst. error) and LIP human parsing (+0.5% mIoU).

Our main contributions are summarized as follows:

- We propose the Lifting Contrastive Learning (LiftedCL) for human-centric pre-training in a simple yet effective way.

- We demonstrate a feasible approach to learn 3D-aware representations via lifting and adversarial learning only using single-view images.

- LiftedCL significantly outperforms state-of-the-art self-supervised learning methods on four human-centric downstream tasks, including 2D and 3D human pose estimation (0.4% mAP and 1.8 mm MPJPE improvement on COCO 2D pose estimation and Human3.6M 3D pose estimation), human shape recovery and human parsing.

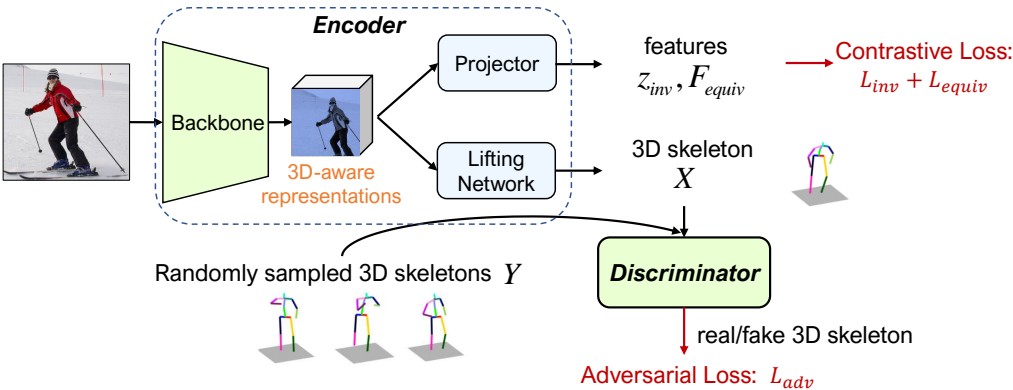

Figure 2: **Overall architecture of the proposed LiftedCL.** The LiftedCL consists of an encoder and a discriminator. The encoder takes an input view to generate invariant feature vector $L_{inv}$, equivariant feature maps $L_{equ}$ and 3D human skeleton $X$. Generated 3D human skeleton is fed into the discriminator to tell whether it is real. Finally, the encoder is optimized by a combination of contrastive loss and adversarial loss.

## 2 RELATED WORKS

**Human-Centric Perception.** With the development of deep learning, human-centric perception has achieved great progress in recent years. Pose estimation, a fundamental yet challenging problem in computer vision, is to localize human anatomical keypoints (e.g., head, shoulder, wrist, etc.) or parts in a given image or 3D space. Lots of work have achieved accurate and robust performance in 2D pose estimation (Xiao et al., 2018; Sun et al., 2019; Yang et al., 2021). For 3D pose estimation (Pavllo et al., 2019; Gong et al., 2021; Wandt & Rosenhahn, 2019), approaches can be roughly grouped into two categories: one-stage approaches which directly learn the 3D poses from images and two-stage approaches which first estimate 2D poses and then lift them to 3D poses. In this work, we focus on the first setting because existing self-supervised learning approaches learn representations from images rather than 2D poses. For 3D human shape recovery (Xu et al., 2021; Choi et al., 2020; Kanazawa et al., 2018), 3D human body is usually represented by a statistical model (e.g. SMPL (Loper et al., 2015)) and the task is to estimate the parameters of the statistical model. Human parsing (Yang et al., 2019; Li et al., 2020; Gong et al., 2018) is to assign each image pixel from the human body to a semantic category. Region analysis and spatial correlation are usually considered to segment body parts. Our goal is to learn human-centric representations for downstream tasks transfer.

**Self-supervised representation learning.** Self-supervised representation learning, a kind of unsupervised learning, has driven significant progress in recent deep learning research. It aims to learn informative and transferable representations for various downstream tasks. Early self-supervised learning approaches explore a wide range of pretext tasks to learn a good representation, including spatial jigsaw puzzles (Noroozi & Favaro, 2016), rotation prediction (Gidaris et al., 2018; Chen et al., 2019), colorization (Zhang et al., 2016) and so on (Doersch et al., 2015; Pathak et al., 2016). However, these approaches achieve very limited success in computer vision and the best-performance approach on a specific task would be sub-optimal on another downstream task.

Recently, contrastive learning has been demonstrated to show incredible promise in computer vision. SimCLR (Chen et al., 2020a) is one breakthrough approach, which adopts the instance discrimination formulation as its pretext task. By applying a diverse set of data augmentations to training images, generated views from the same input are considered as positive pairs while views from different input images as dissimilar pairs. It maximize the similarity in latent space between positive pairs and repelling dissimilar pairs. Besides, MoCo views the contrastive learning as dictionary look-up and introduces momentum update mechanism to ensure the consistency of negative samples. Moreover, some methods (Caron et al., 2018; 2020; 2019) extend instance-level discrimination to predict the cluster assignment. Nonetheless, these approaches are mainly designed for image classification and can be sub-optimal for dense prediction tasks.

To fill the gap between image-level and pixel-level prediction, several approaches (Wang et al., 2021a; Xie et al., 2021; Wang et al., 2021b; O Pinheiro et al., 2020) are proposed to explore dense contrastive representation learning. DenseCL (Wang et al., 2021a) extends and generalizes the

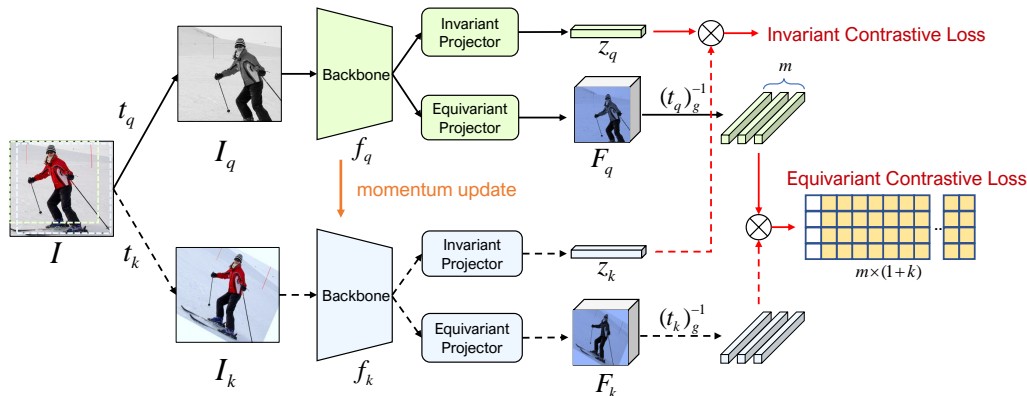

Figure 3: **Contrastive representation learning in our LiftedCL.** We perform invariant and equivariant contrastive learning on 3D-aware representations generated by backbone network.

existing MoCo framework to a dense paradigm. VADeR (O Pinheiro et al., 2020) and PixPro (Xie et al., 2021) map corresponding pixel-wise features in each view to their associated features according to the parameters of the affine transformation. SetSim (Wang et al., 2021b) explore set similarity across views for more robust representation learning. Prior3D (Hou et al., 2021) leverages multi-view geometry information to augment contrastive learning. While these methods achieve promising results on image classification, object detection and semantic segmentation, there still remains untapped potential since 3D human body structure information is neglected during pre-training. HCMoCo (Hong et al., 2022) propose a versatile pre-train model to leverage sparse human body structure priors for human-centric tasks, however, multi-modal data is required during training which is expensive and difficult to collect. In this work, we introduce LiftedCL, a simple yet effective approach for human-centric pre-training which only requires single-view data such as COCO human images.

## 3 METHOD

### 3.1 OVERALL ARCHITECTURE

The overall architecture of our proposed LiftedCL is shown in figure 2, it is composed of an encoder and a discriminator. The encoder consists of a backbone, two projectors and a lifting network. Firstly, given an input view, the representations are extracted by the backbone network, e.g., ResNet (He et al., 2016), and are fed into two projection heads to generate invariant feature vectors $z_{inv}$ and equivariant feature maps $F_{equiv}$ Cheng et al. (2021); Feige (2019). Upon them, an invariant contrastive loss function $L_{inv}$ (He et al., 2020) and an equivariant contrastive loss function $L_{equiv}$ are adopted for representation learning.

Parallel to the projectors, we attach a lifting network to transform the representations into 3D human skeleton. To incorporate 3D human structure information into the representations, improving them to become 3D-aware, a discriminator is trained alternatively to distinguish between lifted 3D skeletons and real 3D skeletons. Note that real 3D skeletons can be randomly sampled using human kinematic priors without any annotated data (see **appendix** B). In all, the whole encoder can be trained by optimizing a combination of contrastive similarity loss and adversarial loss as below.

$$L = \lambda_1(L_{inv} + L_{equiv}) + \lambda_2 L_{adv}, \tag{1}$$

where $\lambda_1$ and $\lambda_2$ controls the relative importance of each loss function. We set $\lambda_1$ and $\lambda_2$ to 0.5.

### 3.2 CONTRASTIVE REPRESENTATIONS

As illustrated in Figure 3, we perform contrastive learning on 3D-aware representations generated by backbone network to obtain good generic representations. The contrastive learning framework can be divided into two parts, invariant contrastive learning and equivariant contrastive learning.

For image-level invariant contrastive learning, we follow (He et al., 2020) to build the learning framework. As shown in Figure 3, given an image $I$, two augmented views $\{I_q, I_k\}$ can be generated by applying various augmentations $t$, where $I_q = t_q(I), I_k = t_k(I), t_q, t_k \in \mathcal{T}$. Each view is first fed

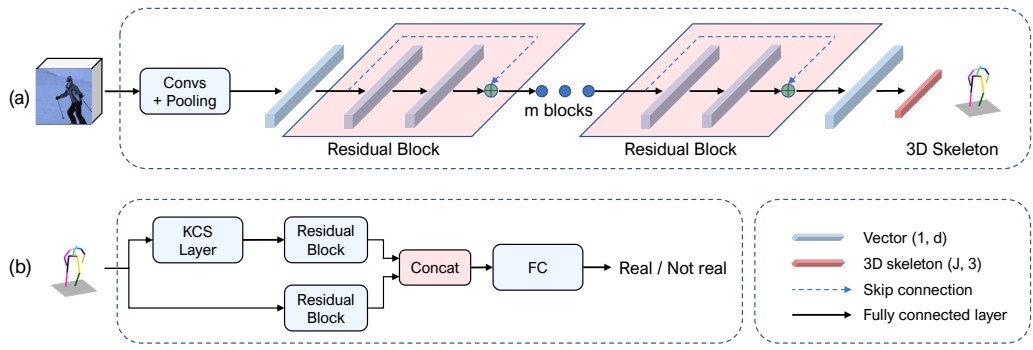

Figure 4: (a) The architecture of lifting network. 3D-aware representations are first vectorized by convolutional and pooling layers and then lifted to 3D by residual blocks and fully connected layers. (b) Network architecture of the discriminator, consisting of a KCS layer and several residual blocks.

into a backbone network $f$ to extract representations and then fed into the invariant projection head to generate a feature vector $z$. Here, $\{I_q, I_k\}$ can be encoded as $\{z_q, z_k\} \in \mathbb{R}^d$ by an encoder and its momentum-updated one. Feature vectors generated from two views of the same input are considered as positive pairs, while negatives are encoded from different images. A standard contrastive loss function is employed to pull $z_q$ close to its positive key $z_{k_+}$ while pushing it away from negative keys,

$$L_{inv} = -log \frac{exp(z_q \cdot z_{k_+}/\tau)}{exp(z_q \cdot z_{k_+}/\tau) + \sum_{z_{k_-}} exp(z_q \cdot z_{k_-}/\tau)}, \qquad (2)$$

where $\tau$ is the temperature hyper-parameter and $z_{k_-}$ comes from a queue.

For equivariant contrastive learning, feature map $F_q \in \mathbb{R}^{C \times S_h \times S_w}$ and its positive pair $F_k$ are generated from two views $I_q, I_k$ of the same input image by a convolutional projection head, then inverse geometric augmentation $(t_q)_g^{-1}, (t_k)_g^{-1}$ is applied separately. Here $(t_q)_g$ is the geometric transformation of $t_q$ and the other is the appearance transformation $(t_q)_a$. Let us define inverse geometric augmented feature maps $\tilde{F}_q = (t_q)_g^{-1}(F_q), \tilde{F}_k = (t_k)_g^{-1}(F_k)$ and they can be seen as $S_h \times S_w$ feature vectors. Let $\tilde{r}^s, \tilde{t}^s$ denotes the $s^{th}$ out of $S_h \times S_w$ encoded vectors from $\tilde{F}_q$ and $\tilde{F}_k$, and here we use $S_h = S_w$ for simpler illustration. If inverse geometric augmentation is not applied before calculating the loss function, the model needs to be invariant to all geometric transformations in $\mathcal{T}$. However, human-centric perception requires equivariance with respect to transformations as these change human poses in augmented views. Moreover, positive pairs can be easily found when applying inverse geometric augmentation as they share the same spatial location. Thus, we define the equivariant loss function as

$$L_{equiv} = \frac{1}{S^2} \sum_s -log \frac{exp(\tilde{r}^s \cdot \tilde{t}_+^s/\tau)}{exp(\tilde{r}^s \cdot \tilde{t}_+^s/\tau) + \sum_{t_-^s} exp(\tilde{r}^s \cdot \tilde{t}_-^s/\tau)}, \qquad (3)$$

where $\tilde{t}_-^s$ is the pooled feature vector of $\tilde{F}_k$ of a view from a different image.

## 3.3 Lifting with Adversarial Learning

Our LiftedCL leverages 3D human body structure information. As shown in Figure 4, a lifting network is employed to transfer representations into 3D human skeleton. The lifting network is mainly composed of several residual blocks consisting of fully connected layers. For a given image $I$, the lifting network extracts a 3D skeleton $\mathbf{X} \in \mathbb{R}^{3 \times J}$ where $J$ stands for the number of joints. Adversarial training is used to guide the lifting network outputting correct 3D skeleton, thus encoding the 3D human structure information into the representations, making it 3D-aware.

The architecture of the discriminator is similar to the lifting network, consisting of fully connected layers, shown in Figure 4. Moreover, to better detect properties of human skeletons such as kinematic chains, bone lengths and joint angle limits, kinematic chain space (KCS) Wandt & Rosenhahn (2019); Wandt et al. (2018) layer is added into our discriminator. KCS can represent joint angles and bone lengths of a human skeleton. We define a bone $b_k$ as the vector between the $r$-th and $t$-th joint.

$$b_k = p_r - p_t = \mathbf{X}c, \qquad (4)$$

where

$$c = (0, \ldots, 0, 1, 0, \ldots, 0, -1, 0, \ldots, 0)^T, \tag{5}$$

with 1 at position $r$ and -1 at position $t$. Then a matrix $\mathbf{B} \in \mathbb{R}^{3 \times b}$ can be calculated by extending $c$ to all bones.

$$\boldsymbol{B} = (b_1, b_2, \ldots, b_b) = \boldsymbol{XC}, \tag{6}$$

here $\boldsymbol{C} \in \mathbb{R}^{J \times b}$, $b$ is the number of bones. By calculating $\boldsymbol{B^T B}$, KCS matrix can be obtained

$$\boldsymbol{\Psi} = \boldsymbol{B^T B} = \begin{pmatrix} l_1^2 & \cdot & \cdot \\ \cdot & \cdot & \cdot \\ \cdot & \cdot & l_b^2 \end{pmatrix}. \tag{7}$$

Each element on its diagonal represents the bone length and other elements represent the relation between two joint vectors. KCS layer helps the discriminator detect 3D human structure information in a more effective way. The complete network is trained alternatively with the discriminator. Let $E$ represent the encoder including the backbone network, projection heads and the lifting network, and $D$ represent the discriminator, then the loss function is

$$L_{adv} = \mathbb{E}_{y \sim p_{data(y)}}[D(y)^2] + \mathbb{E}_{I \sim p_{data(I)}}[(D(E(I)) - 1)^2]. \tag{8}$$

By limiting bone length and joint angle, we randomly sample several rational skeletons $y$. So the whole network can be trained in an unsupervised way under the adversarial learning paradigm.

## 4 EXPERIMENTS

### 4.1 PRE-TRAINING SETUP

Our pre-training experiments are conducted on MS COCO (Lin et al., 2014) and only the training set is used for pre-training. In human-centric perception, human bounding boxes are usually provided, so we crop human regions to better learn representations, resulting $\sim 150K$ images for pre-training. We adopt SGD as the optimizer with initial learning rate of 0.03 and we set its weight decay and momentum to 0.0001 and 0.9. Each pre-training model is optimized on 4 GPUs with a cosine learning rate decay schedule and a mini-batch size of 128 for 200 epochs (details in **appendix C**).

However, most existing state-of-the-art self-supervised learning methods are pre-trained on ImageNet (Deng et al., 2009), so we conduct two-stage pre-training for fair comparison: **Stage I**: The encoder is pre-trained on ImageNet. **Stage II**: The encoder is then pre-trained using COCO human images. For efficiency, we download their official 200-epoch ImageNet pre-trained weights to initialize the encoder as stage I pre-training. For our LiftedCL, we adopt the MoCo-v2 ImageNet 200-epoch pre-trained weights for encoder initialization. Note that MoCo-v2 ImageNet 200-epoch pre-trained weights show inferior performance compared to other SoTA methods, while our LiftedCL still surpasses other methods using it as initialization.

### 4.2 MAIN RESULTS AND DISCUSSION

We compare our LiftedCL with other state-of-the-art self-supervised learning methods on four human-centric tasks, including 2D pose estimation, 3D pose estimation, 3D human shape recovery and human parsing. See more details in **appendix D**.

**2D Human Pose Estimation.** We fine-tune SimplePose (Xiao et al., 2018) detector with ResNet50-C4 backbone on MS COCO (Lin et al., 2014), MPII (Andriluka et al., 2014) and Human3.6M (Ionescu et al., 2013). As illustrated in Table 1, our LiftedCL outperforms the state-of-the-art method PixPro by 0.4% mAP on COCO human pose estimation. Table 2 shows that our LiftedCL surpasses PixPro by 0.4% PCKh@0.5. Fine-tuning results on Human3.6M 2D pose estimation is shown in Table 3, our LiftedCL obtains significant improvement by 0.9% JDR than PixPro.

**3D Human Pose Estimation.** The 3D human pose estimation results on Human3.6M are reported in Table 4. We follow the settings in (Li et al., 2021) to fine-tune a ResNet50 based model. Our LiftedCL outperforms PixPro by significant improvements, 1.8mm and 0.8mm at MPJPE and PA-MPJPE.

Table 1: **2D Pose Estimation fine-tuned on MS COCO.** A ResNet-50 based model Xiao et al. (2018) is adopt for all methods. 'IN' indicates the model pre-trained on ImageNet. 'IN+CC' indicates two-stage pre-training on ImageNet and COCO. '*' indicates using PixPro ImageNet pre-trained weights as stage I pre-training. Each model is pre-trained for 200 epochs.

| pre-train | $mAP \uparrow$ | $AP_{50}$ | $AP^{75}$ | $AP^M$ | $AP^L$ | $AR$ |
|---|---|---|---|---|---|---|
| Random init. | 69.1 | 90.5 | 77.1 | 66.7 | 73.5 | 72.4 |
| Super. IN | 70.4 | 88.6 | 78.3 | 67.1 | 77.2 | 76.3 |
| MoCo-v2 IN Chen et al. (2020b) | 70.3 (-0.1) | 88.6 | 77.9 | 67.0 | 77.1 | 76.2 |
| DenseCL IN Wang et al. (2021a) | 70.6 (+0.2) | 88.7 | 78.2 | 67.4 | 77.3 | 76.4 |
| ReSim-C4 IN Xiao et al. (2021) | 70.4 (+0.0) | 88.5 | 77.9 | 67.1 | 77.2 | 76.3 |
| PixPro IN Xie et al. (2021) | 70.7 (+0.3) | 88.7 | 78.2 | 67.2 | 77.7 | 76.6 |
| MoCo-v2 IN+CC | 70.5 (+0.1) | 88.8 | 78.2 | 67.1 | 77.5 | 76.5 |
| PixPro IN+CC | 70.7 (+0.3) | 89.2 | 78.2 | 67.4 | 77.5 | 76.5 |
| Ours IN+CC | 70.9 (+0.5) | 88.9 | 78.1 | 67.5 | 77.7 | 76.8 |
| Ours* IN+CC | **71.1 (+0.7)** | 89.0 | 78.3 | 67.6 | 77.8 | 76.8 |

Table 2: **2D Pose Estimation fine-tuned on MPII.** (PCKh@0.5)

| pre-train | Hea | Sho | Elb | Wri | Hip | Kne | Ank | Total $\uparrow$ |
|---|---|---|---|---|---|---|---|---|
| Random init. | 96.3 | 94.4 | 87.0 | 80.5 | 86.8 | 81.2 | 76.4 | 86.7 |
| Super. IN | 96.5 | 95.4 | 88.9 | 83.5 | 88.1 | 89.7 | 79.1 | 88.5 |
| MoCo-v2 IN | 96.4 | 95.2 | 89.0 | 83.3 | 87.5 | 83.3 | 79.1 | 88.3 (-0.2) |
| DenseCL IN | 96.8 | 95.4 | 89.0 | 83.7 | 88.1 | 83.9 | 79.6 | 88.6 (+0.1) |
| ReSim-C4 IN | 96.7 | 95.4 | 89.1 | 83.7 | 87.9 | 84.1 | 79.0 | 88.6 (+0.1) |
| PixPro IN | 96.7 | 95.6 | 89.4 | 83.8 | 88.3 | 84.6 | 80.4 | 88.9 (+0.4) |
| MoCo-v2 IN+CC | 96.6 | 95.4 | 88.9 | 84.1 | 88.0 | 83.8 | 79.3 | 88.6 (+0.2) |
| PixPro IN+CC | 96.6 | 95.6 | 89.3 | 83.8 | 88.5 | 84.7 | 81.0 | 89.0 (+0.5) |
| Ours IN+CC | 96.8 | 95.6 | 89.4 | 83.9 | 88.5 | 84.2 | 79.6 | 88.9 (+0.4) |
| Ours* IN+CC | 96.7 | 95.5 | 89.3 | 83.9 | 88.4 | 84.4 | 79.9 | **89.3 (+0.8)** |

**3D Human Shape Recovery.** We follow the settings in (Xu et al., 2021) to fine-tune a ResNet50 based model on a mixed dataset. The 3D human shape recovery results evaluated on 3DPW (von Marcard et al., 2018) are reported in Table 5. MoCo-v2 baseline has no advantage over supervised ImageNet pre-training. Our LiftedCL outperforms state-of-the-art method PixPro by 1.7mm reconstruction error.

**Human Parsing.** The human parsing results on LIP (Liang et al., 2018) are reported in Table 6. A ResNet50 based model with self-correction purification strategy (Li et al., 2020) is fine-tuned. Our LiftedCL outperforms PixPro by 0.1% mACC and 0.5%mIoU.

**Discussion.** LiftedCL achieves significant improvements, especially on Human3.6M 3D pose estimation. When pre-trained on ImageNet, existing self-supervised learning methods do not show superior performance than supervised pre-training except PixPro. Concurrently, when adapting two-stage pre-training, our LiftedCL surpasses PixPro by a large margin, which strongly demonstrates that combining contrastive learning with 3D human prior is beneficial to human-centric tasks.

Besides, self-supervised learning methods bring different performance improvements on downstream tasks, which can be related to the diversity and quantity of fine-tuning data. COCO and MPII consists of a large number of images taken from a wide-range of real-world activities, fine-tuning a pre-trained model on them only achieves limited improvements. For other datasets like Human3.6M or 3DPW, which is collected in a controlled scene or contains a limited number of human subjects, fine-tuning results benefit a lot from pre-training, especially our LiftedCL.

## 4.3 ABLATION STUDY AND VISUALIZATION

We conduct ablation experiments on Human3.6M 2D and 3D pose estimation to show how LiftedCL works. SimplePose (Xiao et al., 2018) and Joint Detection Rate (JDR) is used for 2D human pose estimation, RLEPose (Li et al., 2021) and PA-MPJPE is used for 3D human pose estimation.

Table 3: **2D Pose Estimation fine-tuned on Human3.6M.** (JDR)

| pre-train | root | rhip | Total ↑ |
|---|---|---|---|
| Random init. | 97.1 | 86.4 | 78.7 |
| Super. IN | 94.3 | 89.7 | 85.0 |
| MoCo-v2 IN | 95.8 | 89.6 | 85.6 (+0.6) |
| DenseCL IN | 93.5 | 86.0 | 85.7 (+0.7) |
| ReSim-C4 IN | 96.1 | 90.9 | 86.0 (+1.0) |
| PixPro IN | 96.0 | 91.3 | 86.4 (+1.4) |
| MoCo-v2 IN+CC | 96.2 | 91.8 | 86.2 (+1.2) |
| PixPro IN+CC | 95.9 | 92.0 | 87.1 (+2.1) |
| Ours IN+CC | 95.5 | 91.9 | **88.0 (+3.0)** |

Table 4: **3D Pose Estimation fine-tuned on Human3.6M.** (MPJPE and PA-MPJPE in mm)

| pre-train | MPJPE ↓ | PA-MPJPE ↓ |
|---|---|---|
| Random init. | 70.8 | 52.8 |
| Super. IN | 60.0 | 44.7 |
| MoCo-v2 IN | 61.0 (+1.0) | 46.3 (+1.6) |
| DenseCL IN | 65.1 (+5.1) | 46.7 (+2.0) |
| ReSim-C4 IN | 63.1 (+3.1) | 47.0 (+2.3) |
| PixPro IN | 58.9 (-1.1) | 43.7 (-1.0) |
| MoCo-v2 IN+CC | 61.2 (+1.2) | 45.4 (+0.7) |
| PixPro IN+CC | 58.7 (-1.3) | 43.3 (-1.4) |
| Ours IN+CC | **56.9 (-3.1)** | **42.5 (-2.2)** |

Table 5: **Human shape recovery fine-tuned on 3DPW.**

| pre-train | Reconst. error ↓ |
|---|---|
| Random init. | 116.3 |
| Super. IN | 106.1 |
| MoCo-v2 IN+CC | 111.5 (+5.4) |
| PixPro IN+CC | 104.5 (-1.6) |
| Ours IN+CC | **102.8 (-3.3)** |

Table 6: **Human parsing fine-tuned on LIP.**

| pre-train | $mACC$ ↑ | $mIoU$ ↑ |
|---|---|---|
| Random init. | 63.5 | 50.3 |
| Super. IN | 63.7 | 51.8 |
| MoCo-v2 IN+CC | 64.9 (+1.2) | 53.0 (+1.2) |
| PixPro IN+CC | 67.5 (+3.8) | 54.6 (+2.8) |
| Ours IN+CC | **67.6 (+3.9)** | **55.1 (+3.3)** |

**Pre-train dataset.** In Table 7, we show the results of different pre-training datasets. When only using Human3.6M, LiftedCL improves the performance by 2.7% JDR and 4.9 mm PA-MPJPE compared to random initialization. Using COCO brings 6.9% JDR and 8.4 mm PA-MPJPE improvements, which is superior than supervised ImageNet pre-training. Note that COCO pre-training only uses $\sim 1/7$ images and less iterations. As for the results of two-stage pre-training, it surpasses the ImageNet supervised pre-training method by 3.0% JDR and 2.2 mm PA-MPJPE.

**Loss function.** Our loss function is composed of three parts: $L_{inv}$, $L_{equiv}$ and $L_{adv}$. To show how each component contributes to LiftedCL, we pre-train the backbone using different loss combinations. We report the results in Table 8. We refer to the ImageNet pre-training method as our baseline. It shows that invariant contrastive loss improves the baseline by 1.2% JDR and -0.7 mm PA-MPJPE. Adding the equivariant loss brings another 0.9% JDR and 1.1 mm PA-MPJPE gains. When all three losses are used, the pose estimation performance improves for 3.0% JDR and 2.2 mm PA-MPJPE compared to the baseline. Meanwhile, directly applying the lifting network does not bring notable improvements. It may be because lifting network would collapse without good representations learned by contrastive learning.

Table 7: **Ablation study of pre-training dataset.**

| Pre-train method | Pre-train dataset | epoch | JDR ↑ | PA-MPJPE ↓ |
|---|---|---|---|---|
| Random init. | - | - | 78.7 | 52.8 |
| Supervise | ImageNet | 100 | 85.0 | 44.7 |
| MoCo-v2 | H36M | 200 | 79.8 (-5.2) | 49.6 (+4.9) |
| MoCo-v2 | COCO | 200 | 83.6 (-1.4) | 45.9 (+1.2) |
| MoCo-v2 | ImageNet | 200 | 85.6 (+0.6) | 46.3 (+1.6) |
| MoCo-v2 | ImageNet + COCO | 200 | 86.2 (+1.2) | 45.4 (+0.7) |
| Ours | H36M | 200 | 81.4 (-3.6) | 47.9 (+3.2) |
| Ours | COCO | 200 | 85.6 (+0.6) | 44.4 (-0.3) |
| Ours | ImageNet + COCO | 200 | 88.0 (+3.0) | 42.5 (-2.2) |

Table 8: **Ablation study of different loss functions.**

| Method | JDR ↑ | PA-MPJPE ↓ |
|---|---|---|
| Supervise IN | 85.0 | 44.7 |
| Inv. | 86.2 | 45.4 |
| Inv. + Equiv. | 86.9 | 44.3 |
| Lifting | 85.2 | 44.8 |
| Inv. + Equiv. + Lifting | 88.0 | 42.5 |

Table 9: **Ablation study of fine-tuning w/o 3D prior.**

| Method | JDR ↑ | PA-MPJPE ↓ |
|---|---|---|
| Supervise IN | 85.0 | 44.7 |
| + prior | 86.1 | 44.2 |
| Ours | 88.0 | 42.5 |
| + prior | 88.7 | 42.3 |

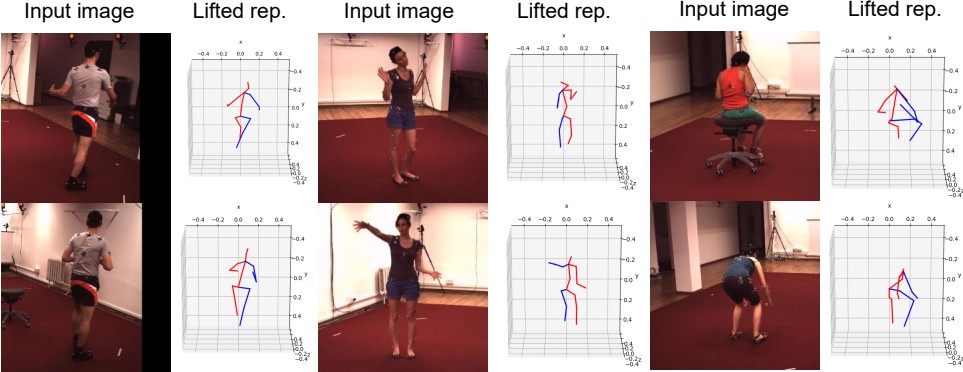

Figure 5: **visualization of 3D skeletons generated by lifting network.**

**3D prior.** We further investigate the effect of 3D prior. When fine-tuning with 3D human kinematic prior, we additionally train a lifting network to estimate 3D pose. Based on the estimated 3D pose, the bone length and joint ankles can be calculated and if they are not within the reasonable range, an extra loss will be added to penalize the incorrect estimation. As shown in Table 9, fine-tuning with 3D human kinetic prior improves the JDR by 1.1% and PA-MPJPE by 0.5 mm, demonstrating the effectiveness of 3D human kinetic prior. However, even when fine-tuning without 3D prior, our LiftedCL surpassed that by 1.9% JDR and 1.8 mm PA-MPJPE, which shows that our LiftedCL can leverage 3D human kinetic prior more effectively. Moreover, our LiftedCL can also benefit from 3D prior when fine-tuning, which is in accordance with our intention that LiftedCL is designed for pre-training and 3D prior can help on both pre-training and fine-tuning stage.

**3D skeleton visualization.** We visualize the 3D skeletons generated by lifting network in Figure 5. Given an input view, generated 3D skeletons can roughly reveal the human skeleton. It is in accordance with our intention that human body structure information is learned during pre-training. Meanwhile, the generated 3D skeletons is not very accurate as no annotated data is used and only an adversarial loss is applied for 3D-aware human-centric representation learning.

## 5 CONCLUSION

In this work, we proposed a simple yet effective self-supervised representation learning framework LiftedCL for 3D-aware human-centric pre-training. By lifting 2D contrastive representations to 3D human skeleton and adapt adversarial learning, LiftedCL encodes 3D human structure information into the learned representations. Besides, with 3D human kinematic prior, real 3D skeletons can be generated by randomly sampling reasonable 3D human skeletons, which alleviates the problem of expensive and time-consuming data collection. Our method shows promising improvements in a serious of human-centric tasks, and more importantly, shows the great potential of improving contrastive learning with 3D prior through adversarial learning. Our limitation is that cropped human images are required during pre-training. In the future, we will further investigate how to embed 3D prior into representations with single-view input on object detection.

## 6 ACKNOWLEDGMENTS

This work was supported by the National Natural Science Foundation of China under Nos. 62276061.

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

## A  GENERATION OF COCO HUMAN IMAGES

Our pre-training experiments on COCO only use human images, which are generated by cropping human regions according to the annotations. For each human bounding box, the box is first extended to a fixed aspect ratio, i.e., height : width = 4 : 3, and then we crop from images without distorting human bounding boxes' aspect ratio. Finally, we resize the cropped image to a fixed size, $256 \times 192$. Figure 6 shows some examples of our cropped human images.

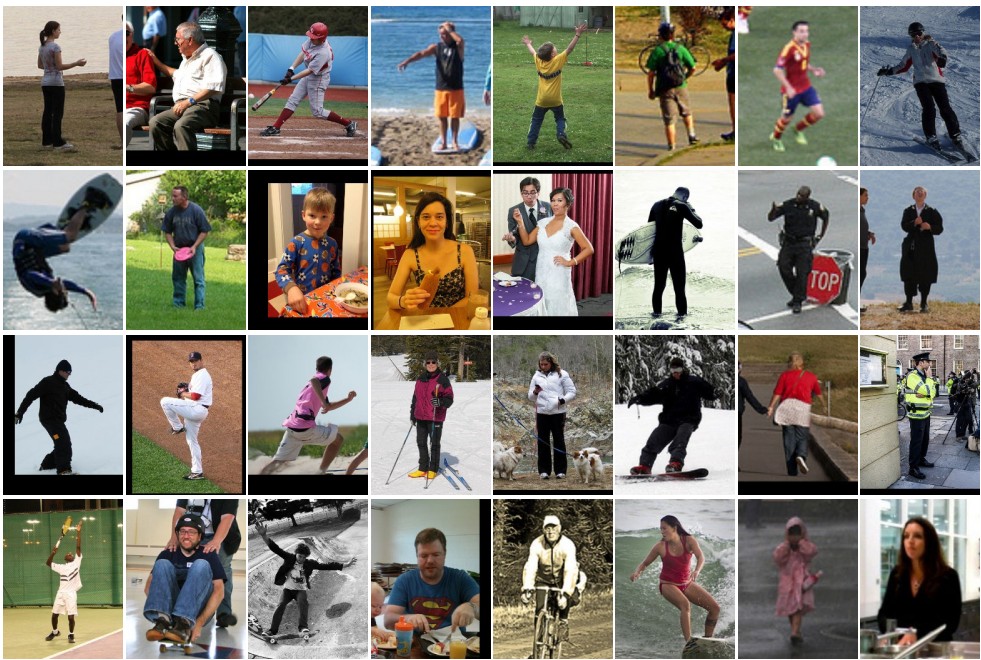

Figure 6: **Cropped human images from MS COCO.**

## B  SAMPLING OF REAL 3D SKELETONS

We generate a set of real 3D samples for adversarial learning by randomly sampling reasonable 3D human skeletons satisfying 3D human kinematic prior without any annotated data. We define a 3D human skeleton

$$\mathbf{P} = (j_{root}, j_{right\_pelvis}, j_{right\_knee}, \cdots, j_{left\_elbow}, j_{left\_wrist}) \in \mathbb{R}^{3 \times J}, \tag{9}$$

where $J$ stands for the number of joints and we use a 17 joint human skeleton. We first set root joint $j_{root} = (0, 0, 0)^T$ and then any other joint can be generated using its parent joint and a bone vector.

$$j_{child} = j_{parent} + \boldsymbol{R} \cdot \alpha \cdot b_{parent\_child}. \tag{10}$$

$j_{child}$ and $j_{parent}$ represent the child joint and its parent joint separately. $b_{pc} \in \mathbb{R}^3$ is the default bone vector related to these two joints. We first define a template 3D skeleton. In this way, each default bone vector $b_{pc}$ can be obtained according to that. $\alpha$ is the random length ratio and $\boldsymbol{R}$ is the random rotation matrix to change the bone length and direction. We set $\alpha \in [0.9, 1.1]$ for all bones and set different rotation ranges for different bones according to the human kinematic prior. For example, for the knee joint, $b_{pelvis\_knee} = (0, 45, 0)^T$, $\alpha \in [0.9, 1.1]$ and $\boldsymbol{R} = euler(a, b, c), a \in [-90°, 90°], b \in [-180°, 180°], c \in [-90°, 90°]$.

The 3D human skeleton can be generated by uniformly sampling $\alpha$ and $\boldsymbol{R}$ in their ranges. For the generated 3D skeleton, it is scaled by dividing it by its Frobenius norm. To simulate different camera

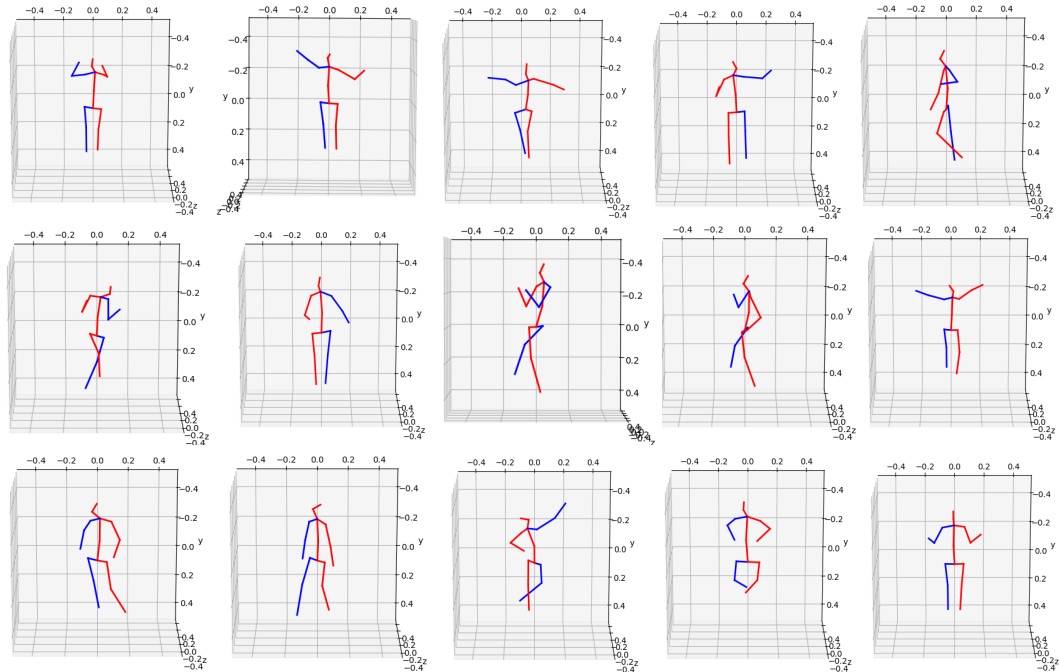

Figure 7: **Randomly sampled 3D human skeletons.**

views, generated 3D skeletons are randomly rotated around $Z$ axis.We generate a set of $\sim 10K$ 3D human skeletons as real 3D samples for adversarial learning in our LiftedCL. Figure 7 shows some examples of our randomly sampled 3D skeletons.

## C    IMPLEMENTATION DETAILS OF LIFTEDCL

We adopt ResNet50 (He et al., 2016) as our default backbone, it is worth noting that our approach can be employed to other networks, e.g. HRNet (Sun et al., 2019). Following (Chen et al., 2020a; He et al., 2020), invariant projection head consists of a global average pooling layer and a two layer MLP, which takes the feature maps as input and generates a global feature vector for each view. For equivariant projection head, it outputs features maps having the same size of the input consisting of two $1 \times 1$ convolution layers with a ReLU layer between them. The first convolution layer's dimension is 2048, and the final output dimension is 128. For both the invariant and equivariant contrastive learning, the dictionary size is set to 16384. The momentum is set to 0.999. Shuffling BN (He et al., 2020) is used during training. The temperature $\tau$ in contrastive loss is set to 0.2.

The data augmentation pipeline consists of $256 \times 256$-pixel random resized cropping, random color jittering, random gray-scale conversion, gaussian blurring and random horizontal flip following (Chen et al., 2020b). Rotation is not used for fair comparison with previous self-supervised methods. Inverse geometry transformations are applied in equivariant contrastive loss, the output feature maps will be flipped horizontally if the input view was flipped before. For random resized cropping, shared area in two views' feature maps $F_q, F_k$ will be cropped and resized to $16 \times 16$ resolution as inverse geometric augmented feature maps, then equivariant contrastive loss can be calculated. The training code of our framework is modified from the official PyTorch implementation of MoCo[1].

---

[1]https://github.com/facebookresearch/moco

## D  FINE-TUNING DETAILS

**2D Human Pose Estimation.** We fine-tune SimplePose (Xiao et al., 2018) detector with ResNet50-C4 backbone on MS COCO (Lin et al., 2014), MPII (Andriluka et al., 2014) and Human3.6M (Ionescu et al., 2013). The learning schedule follows the setting (Xiao et al., 2018).

For 2D human pose estimation on COCO, we adapt the official codes of SimplePose[2]. The fine-tuning is conducted on COCO $train2017$ dataset for 140 epochs with a mini-batch size of 128, including $57K$ images and $150K$ person instances. The evaluation is conducted on the $val2017$ set, containing 5000 images. The standard evaluation metric is based on Object Keypoint Similarity (OKS). We follow the evaluation metric in (Sun et al., 2019) and report standard average precision and recall scores: $AP^{50}$ ($AP$ at $OKS = 0.50$), $AP^{75}$, $AP$ (the mean of $AP$ scores at 10 positions, $OKS = 0.50, 0.55, \ldots, 0.90, 0.95$), $AP^M$ for medium objects, $AP^L$ for large objects, and $AR$ at $OKS = 0.50, 0.55, \ldots, 0.90, 0.95$.

When fine-tuning SimplePose on MPII Human Pose dataset, we also use its official codes. MPII consists of $\sim 28K$ subjects for training and $\sim 12K$ subjects for testing. The standard metric PCKh@0.5 ($\alpha = 0.5$) score (head-normalized probability of correct keypoint) score is used.

For Human3.6M 2D pose estimation, we refer to the official codes in (Qiu et al., 2019)[3] and follow its default training schedule. Human3.6M dataset contains $\sim 63K$ images for training and $\sim 9K$ images for testing. The pose estimation accuracy is measured by Joint Detection Rate (JDR), which means the percentage of the successfully detected joints. The estimated joint is regarded as successfully detected if the distance between it and the groundtruth location is smaller than a half of the head size.

**3D Human Pose Estimation.** For 3D human pose estimation in Human3.6M, we fine-tune a model with the ResNet50 backbone (Li et al., 2021) using its default training schedule[4] on subjects S1, 5, 6, 7, 8 and evaluate on subjects S9 and S11. We use two evaluation metrics: Mean Per Joint Position Error (MPJPE) in millimeters and MPJPE over aligned predictions with GT 3D poses by a rigid transformation (PA-MPJPE).

**3D Human Shape Recovery.** We follow the training schedule and settings in (Xu et al., 2021)[5] to fine-tune a ResNet50 based model on a mixed dataset. The 3D human shape recovery results are evaluated on 3DPW (von Marcard et al., 2018) and reconstruction error is used to evaluate the performance.

**Human Parsing.** A ResNet50 based model with self-correction purification strategy (Li et al., 2020)[6] is fine-tuned on LIP dataset. LIP dataset contains 50462 images with pixel-wise annotations with 19 semantic human part labels. We fine-tune pre-trained models on $train$ set and evaluate the performance on $validation$ set, consisting 30462 images and 10000 images respectively.

## E  ADDITIONAL VISUALIZATION OF LIFTED 3D SKELETONS

Additional visualization of lifted 3D skeletons by lifting network is shown in Figure 8.

---

[2]https://github.com/leoxiaobin/deep-high-resolution-net.pytorch

[3]https://github.com/microsoft/multiview-human-pose-estimation-pytorch

[4]https://github.com/Jeff-sjtu/res-loglikelihood-regression

[5]https://github.com/xuxy09/RSC-Net

[6]https://github.com/GoGoDuck912/Self-Correction-Human-Parsing

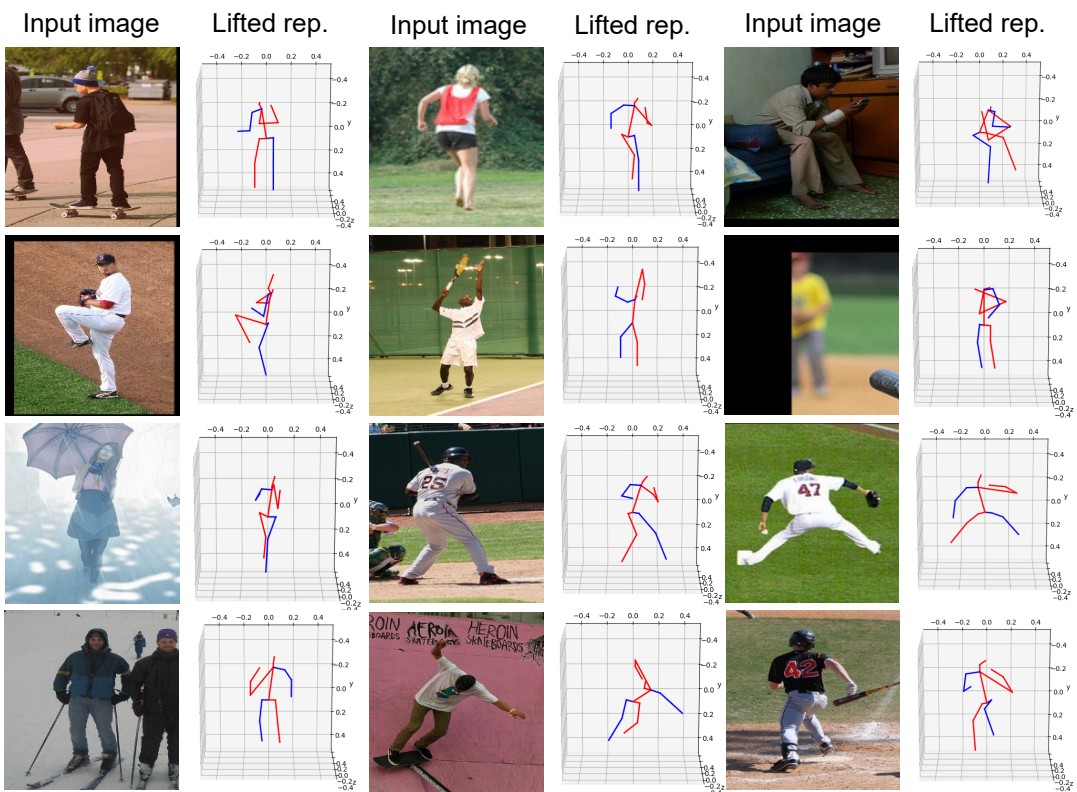

Figure 8: **Additional visualization of lifted 3D skeletons.**

