# OpenReview forum: "LiftedCL: Lifting Contrastive Learning for Human-Centric Perception"
_ICLR.cc/2023/Conference — ICLR 2023 poster_

### Official Review · Reviewer_1MfS · 2022-10-24

**Confidence:** 3
**Clarity, Quality, Novelty And Reproducibility:** 1. Clarity is excellent. Paper is wel…
**Correctness:** 4
**Technical Novelty And Significance:** 2
**Empirical Novelty And Significance:** 3
**Recommendation:** 8

**Strength And Weaknesses:**

Strengths
1. Paper's motivations and methods are well presented and supported by experimental results.
2. Experimental setups are comprehensive and strongly supports the core claim of the paper: self-supervised learning network with contrastive approach to generate 3D representation of human images is superior to other self-supervised approaches.
3. Ablation studies are thorough and well-done.

Weaknesses
1. Incremental novelty as proposed method is similar to other approaches for CV tasks, including image classification, object detection, semantic segmentation, etc
2. Experimental results only compare with other self-supervised methods.

**Summary Of The Paper:**

Paper proposed a novel architecture for self-learning of human-centric perception problems: 2D/3D human pose estimation, human shape recovery and human parsing. The proposed method is simple and effective -- exceeding SOTA for several benchmarks. The encoder subnetwork uses Contrastive Learning and Lifting network to generate the 3D skeleton of the input; this is then combined with Adversarial loss from Discriminator sub-network to improve the encoder.



**Summary Of The Review:**

Overall, paper is a good piece of work which should be accepted for ICLR. There are aspects of the work which are incremental, e.g. using of Contrastive Loss, 3D structure representation of human. However, the scientific contributions of it are still sufficiently significant, especially in view of its strong empirical experimental results.

---

> ### Author Response · Authors · 2022-11-08
> **Response to Reviewer 1MfS**
>
> We sincerely appreciate your careful and thoughtful comments and time. We will explain your concerns point by point.
>
> **Q1: Incremental novelty**
>
> **A1:** We appreciate your objective comment on our novelty. We believe "a new use of an existing technique with different goals" is meaningful when the new use is effective and brings a new perspective. To the best of our knowledge, we are the first learn 3D-aware human-centric representations only using single-view images, while previous works may need multi-view data [1] or multi-modality data [2]. Experimental results show the effectiveness of our LiftedCL. Moreover, our work could be beneficial to the general community as it shows the feasibility of learning 3D-aware representations via lifting and adversarial learning. Our idea has the potential to be applied in other tasks, e.g., object detection. We hope our work can inspire others when learning 3D-aware representations.
>
> **Q2: Experimental results only compare with other self-supervised methods.**
>
> **A2:** We follow previous works, MoCo [3], SimCLR [4], DenseCL [5] and PixPro [6], to compare our method only with other self-supervised learning methods. We all focus on pre-training stage, so we pre-train the backbone (e.g., ResNet-50) using different SSL methods and fine-tune using the same downstream task model. The used downstream task model can be chosen in a wide range of existing SOTA models. We are not sure if our explanation could solve your concerns, if not, we may have to bother you to comment and then we shall know.
>
> **Q3: Reproducibility and codes**
>
> **A3:** Training codes and our pre-trained models will be released in GitHub.
>
> **References**
>
> [1] Ji Hou, Saining Xie, Benjamin Graham, Angela Dai, and Matthias Nießner. Pri3d: Can 3d priors help 2d representation learning?. In Proceedings of the IEEE/CVF International Conference on Computer Vision, pages 5693-5702, 2021.
>
> [2] Fangzhou Hong, Liang Pan, Zhongang Cai, and Ziwei Liu. Versatile Multi-Modal Pre-Training for Human-Centric Perception. In Proceedings of the IEEE/CVF Conference on Computer Vision and Pattern Recognition, pages 16156-16166, 2022.
>
> [3] Kaiming He, Haoqi Fan, Yuxin Wu, Saining Xie, and Ross Girshick. Momentum contrast for unsupervised visual representation learning. In Proceedings of the IEEE/CVF conference on computer vision and pattern recognition, pages 9729–9738, 2020.
>
> [4] Ting Chen, Simon Kornblith, Mohammad Norouzi, and Geoffrey Hinton. A simple framework for contrastive learning of visual representations. In International conference on machine learning, pages 1597–1607. PMLR, 2020.
>
> [5] Xinlong Wang, Rufeng Zhang, Chunhua Shen, Tao Kong, and Lei Li. Dense contrastive learning for self-supervised visual pre-training. In Proceedings of the IEEE/CVF Conference on Computer Vision and Pattern Recognition, pages 3024–3033, 2021.
>
> [6] Zhenda Xie, Yutong Lin, Zheng Zhang, Yue Cao, Stephen Lin, and Han Hu. Propagate yourself: Exploring pixel-level consistency for unsupervised visual representation learning. In Proceedings of the IEEE/CVF Conference on Computer Vision and Pattern Recognition, pages 16684–16693, 2021.

---

### Official Review · Reviewer_3nTM · 2022-10-24

**Confidence:** 3
**Correctness:** 4
**Technical Novelty And Significance:** 2
**Empirical Novelty And Significance:** 2
**Recommendation:** 5

**Clarity, Quality, Novelty And Reproducibility:**

The paper is well written and easy to follow. I have concerns about originality as the key contributions of this work seem a bit derivative of existing works, equivariance and invariance  formulation is very similar to Cheng et al. ICCV'21 and lifting to 3D using KCS is similar to RepNet, CVPR'19. Can the author clarify why the current approach is not a combination of these two?

**Details Of Ethics Concerns:**

I do not see immediate ethical concerns for this work.

**Strength And Weaknesses:**

Strengths:
+ Contrastive learning for 3D tasks is quite useful because acquiring annotated 3D data is very hard for for a lot of tasks. Unsupervised feature learning approaches make a lot of sense.
+ I like the ablation study to motivate the choice of using invariant and equivariant feature learning.
+ The approach is straightforward and easy to follow.

Weakness:
- The idea of using equivariance and invariance for feature learning has been proposed in Cheng et al. ICCV'21 (not cited). Eg: They also use contrastive learning on augmented images and use that augmentation of the same image are positive pairs and different images serve as negative pairs. How is the proposed formulation different?

- How is KCS + adversarial learning (Sec 3.3) different than Critic Network in RepNet? Eq. 4,5,6,7 here are same as Eq. 3,4,5,6,7 in RepNet.

- Sec 3.2: To learn equivariant features, since the inverse transformation is known, why do we need to use contrastive learning? We have direct supervision that features of positive pairs should be the same after the inverse transform. Can the authors comment on this?

- Datasets like H36m are a bit saturated for pose estimation as they are a bit synthetic and controlled. 3DPW is more relevant dataset these days. Authors use it for shape recovery but not pose estimation evaluation. Can the authors comment on this?

Clarifications and minor suggestions:
- Sec 3.2: What augmentations are used?
- Add citation to Wandt et al. ECCVW'18 and RepNet, CVPR'19 in Sec 3.3. Authors mention this in related work but it is better to add the citation here as well, otherwise KCS looks like a contribution of this work.
- PixPro is just an unsupervised feature representation method. How do the authors use it for 2D pose prediction in Sec 4.2? Can the authors provide more details?
- Add a bullet list of key contributions. It clearly distinguishes the contributions of this work from the other works. This is quite useful as the final approach often uses components from other works.
- The ideas of equivariance and invariance are quite well known in the 3D vision community. Authors might be interested in checking some of these works out. eg: ART, Zhou et al. 3DV'22.

**Summary Of The Paper:**

The authors propose, Lifting Contrastive Learning (LiftedCL), a feature representation learning approach based on contrastive learning with primary focus on the task of 2D/3D human pose estimation from a single image. Authors further use kinematic chain space (KCS) layer and a discriminator to regularise the skeleton. The key idea is to decompose the information into two, invariant and equivariant features.

Authors provide key experiments on MSCOCO, MPII and H36m datasets. Ablation study shows that the unsupervised pre-training with invariant and equivariant features improves performance over vanilla supervised training.

**Summary Of The Review:**

I'm currently on the fence for this work. The key ideas seem to be coming from other works (see weakness section) and I've seen better numbers on 2D/3D pose estimation task. Although these SOTA works (including the best performing version of SimplePose, whose smaller version authors use as baseline) use much bigger networks than Res50 so they are not directly comparable.
I don't see a very compelling reason to accept this work. I'm open to changing my mind if authors can clarify the limited technical novelty aspect.

---

> ### Author Response · Authors · 2022-11-08
> **Response to Reviewer 3nTM (Part 2)**
>
> **Q4: Clarifications**
>
> **A4:**
>
> **1. Sec 3.2: What augmentations are used?**
>
> We mention this in Appendix C, implementation details of LiftedCL, "The data augmentation pipeline consists of 256×256-pixel random resized cropping, random color jittering, random gray-scale conversion, gaussian blurring and random horizontal flip following (Chen et al., 2020b). Rotation is not used for fair comparison with previous self-supervised learning methods."
>
> **2. citations**
>
> We will add citation of KCS [9] and RepNet [10] in the method section to avoid confusion. We will also add citation of "Cheng et al. ICCV'21" [6] and "Feige et al. ICML'19" [7] in the method section.
>
> **3. PixPro is just an unsupervised feature representation method. How do the authors use it for 2D pose prediction in Sec 4.2?**
>
> Here is detailed explanation, our LiftedCL and compared SSL methods focus on pre-training the backbone, e.g., ResNet-50. When evaluating one SSL method, we first use this method to pre-train the backbone, and then we use the pre-trained weights to initialize the the backbone of specific downstream task model. In the end, the downstream task model is fine-tuned on its downstream task dataset.
>
> **4. Add a bullet list of key contributions.**
>
> Our main contributions are summarized as follows:
>
> 1. We propose the Lifting Contrastive Learning (LiftedCL) for human-centric pre-training in a simple yet effective way.
>
> 2. We demonstrate a feasible approach to learn 3D-aware representations via lifting and adversarial learning only using single-view images.
>
> 3. LiftedCL significantly outperforms state-of-the-art self-supervised learning methods on four human-centric downstream tasks, including 2D and 3D human pose estimation (0.4% mAP and 1.8 mm MPJPE improvement on COCO 2D pose estimation and Human3.6M 3D pose estimation), human shape recovery and human parsing.
>
> We will add this in our final version.
>
>
> **References**
>
> [1] Kaiming He, Haoqi Fan, Yuxin Wu, Saining Xie, and Ross Girshick. Momentum contrast for unsupervised visual representation learning. In Proceedings of the IEEE/CVF conference on computer vision and pattern recognition, pages 9729–9738, 2020.
>
> [2] Ting Chen, Simon Kornblith, Mohammad Norouzi, and Geoffrey Hinton. A simple framework for contrastive learning of visual representations. In International conference on machine learning, pages 1597–1607. PMLR, 2020.
>
> [3] Zhenda Xie, Yutong Lin, Zheng Zhang, Yue Cao, Stephen Lin, and Han Hu. Propagate yourself: Exploring pixel-level consistency for unsupervised visual representation learning. In Proceedings of the IEEE/CVF Conference on Computer Vision and Pattern Recognition, pages 16684–16693, 2021.
>
> [4] Hou Ji, Saining Xie, Benjamin Graham, Angela Dai, and Matthias Nießner. Pri3d: Can 3d priors help 2d representation learning?. In Proceedings of the IEEE/CVF International Conference on Computer Vision, pages 5693-5702, 2021.
>
> [5] Fangzhou Hong, Liang Pan, Zhongang Cai, and Ziwei Liu. Versatile Multi-Modal Pre-Training for Human-Centric Perception. In Proceedings of the IEEE/CVF Conference on Computer Vision and Pattern Recognition, pages 16156-16166, 2022.
>
> [6] Zezhou Cheng, Jong-Chyi Su, and Subhransu Maji. On equivariant and invariant learning of object landmark representations. In Proceedings of the IEEE/CVF International Conference on Computer Vision, pages 9897-9906, 2021.
>
> [7] Ilya Feige. Invariant-equivariant representation learning for multi-class data. In International Conference on Machine Learning, pages 1882-1891, 2019.
>
> [8] Julieta Martinez, Rayat Hossain, Javier Romero, and James J Little. A simple yet effective baseline for 3d human pose estimation. In Proceedings of the IEEE international conference on computer vision, pages 2640–2649, 2017.
>
> [9] Bastian Wandt, Hanno Ackermann, and Bodo Rosenhahn. A kinematic chain space for monocular motion capture. In Proceedings of the European Conference on Computer Vision (ECCV) Workshops, pages 0-0, 2018.
>
> [10] Bastian Wandt, and Bodo Rosenhahn. Repnet: Weakly supervised training of an adversarial reprojection network for 3d human pose estimation. In Proceedings of the IEEE/CVF Conference on Computer Vision and Pattern Recognition, pages 7782-7791, 2019.
>
> [11] Xinlong Wang, Rufeng Zhang, Chunhua Shen, Tao Kong, and Lei Li. Dense contrastive learning for self-supervised visual pre-training. In Proceedings of the IEEE/CVF Conference on Computer Vision and Pattern Recognition, pages 3024–3033, 2021.
>
> [12] Jiefeng Li, Siyuan Bian, Ailing Zeng, Can Wang, Bo Pang, Wentao Liu, and Cewu Lu. Human pose regression with residual log-likelihood estimation. In Proceedings of the IEEE/CVF International Conference on Computer Vision, pages 11025-11034, 2021.

---

> ### Author Response · Authors · 2022-11-08
> **Response to Reviewer 3nTM (Part 1)**
>
> We sincerely appreciate your careful and thoughtful comments and time. We will explain your concerns point by point.
>
> **Q1: Novelty**
>
> **A1:** The core idea of our work is to learn 3D-aware human-centric representations which can be transferred to various downstream tasks. However, the learned representations using previous self-supervised learning methods are not 3D-aware [1, 2, 3], or the pre-training process requires multi-view data or multi-modality data [4, 5]. Our main contribution is that we show a feasible way to learn 3D-aware representations for human-centric tasks via lifting and adversarial training only using single-view images. Moreover, our work could also inspire other works, e.g., learning 3D-aware representations for object detection.
>
> We believe "a new use of an existing technique with different goals" is meaningful when the new use is effective and brings a new perspective. Although invariance and equivariance [3, 6, 7], lifting [8] and KCS [9, 10] have been proposed in previous works, we first use them for 3D-aware human-centric representation learning and show promising results. Our work focus on pre-training the backbone and we hope our work can inspire others when learning 3D-aware representations.
>
> More specifically, In our work, invariant and equivariant contrastive learning is to help learn generic representations, experimental results in Table 8 also show that this part is essential for the final 3D-aware representation learning. However, how to encode 3D human information into the representations is not solved. Inspired by previous 3D pose estimation works [8, 10], we lift learned representations to 3D skeleton format and use adversarial learning to regularize them, thus encoding 3D human information into the learned representations, making them 3D-aware. The main difference between our LiftedCL and RepNet is that we focus on learning 3D-aware representations for human-centric pre-training, while RepNet focuses on fine-tuning stage.
>
> **Q2. Why do we need to use contrastive learning to learn equivariant features**
>
> **A2:** We follow previous works [3, 11] to conduct contrastive learning when learning equivariant features. To this end, our proposed LiftedCL can be compared with other SSL methods in a fair way.
>
> **Q3. 3DPW dataset not used for pose estimation**
>
> **A3:** When comparing our proposed LiftedCL with other SSL methods on 3D human pose estimation, we choose [12] to fine-tune as this model uses a ResNet-50 backbone which can be pre-trained using our and other SSL methods. We directly refer to its official codes for convenience, however, it only conducts experiments on Human3.6M dataset, so we do not test our LiftedCL in 3DPW 3D pose estimation.

---

### Official Review · Reviewer_5VTe · 2022-10-25

**Confidence:** 4
**Correctness:** 4
**Technical Novelty And Significance:** 3
**Empirical Novelty And Significance:** Not applicable
**Recommendation:** 8

**Clarity, Quality, Novelty And Reproducibility:**

### Clarity
The paper is straightforward and easy to follow.

### Quality
The quality of the work is good.

### Novelty
I think the idea of encoding 3D-aware representations in human-centric tasks via lifting-to-3D contrastive learning is novel and it can inspire future studies in this direction.

### Reproducibility
There seems to be a sufficient amount of detail in the paper for reproducibility. It will be better if the authors plan to release their code. It is not currently mentioned in the submission.

**Strength And Weaknesses:**

### Strength
- The intuition of learning a robust representation that encodes 3D human information makes a lot of sense to me.

- The designed pipeline is straightforward. It could inspire further development in this direction.

- The invariance and equivariance training is novel in the context of 3D human feature learning tasks.

- I like the idea of having unpaired randomly sampled 3D skeletons as auxiliary training data.

- Experiments show the solid performance of the proposed method when trained on ImageNet + COCO: downstream tasks have significant improvement. That validates the idea of encoding 3D human structure information is useful for human-centric tasks.

### Weakness
- I think the paper could use a strong baseline in which all the 3D elements are replaced by 2D counterparts. For example, instead of lifting 2D features to 3D skeletons, the network will output 2D skeletons and a discriminator will compare those with randomly sampled real 2D skeletons. Other parts of the network will be kept as much as possible. If the proposed LiftedCL outperforms this 2D alternative on 2D human tasks such as pose estimation and parsing, then it is clear that 3D is essential to both 2D and 3D human tasks. I agree with the authors that 3D is important but it will be more convincing with such an experiment.

- I might have missed it, but is there an ablation study on the effectiveness of the KCS layer?


**Summary Of The Paper:**

This paper proposes the Lifting Contrastive Learning (LiftedCL) workflow to train robust human-centric representations that are useful for downstream 2D and 3D human tasks such as pose estimation, shape prediction, and parsing. There are two core components. The contrastive learning part generates features that are invariant and equivariant (under inverse transforms). The "lifting" part generates 3D skeletons from the human-centric representations and is trained with a discriminator along with randomly sampled real 3D skeletons. Experiments show that the proposed training produces more useful features for downstream tasks than competitive alternatives such as MoCo-v2 and PixPro.

**Summary Of The Review:**

Overall, I like this paper because it studies an interesting and important problem of encoding 3D-aware representations in human-centric tasks and provides a viable solution. The design components in this workflow could inspire future research in this direction. The experiment results are solid on multiple human-centric tasks. The paper could be further strengthened by adding the 2D baseline mentioned above, but it is a good contribution to the community in its current form.

---

> ### Author Response · Authors · 2022-11-08
> **Response to Reviewer 5VTe**
>
> We sincerely appreciate your careful and thoughtful comments and time, especially your approval for our idea of encoding 3D information into representations, which really means a lot for us. We will explain your concerns point by point.
>
> **Q1: A strong baseline in which all the 3D elements are replaced by 2D counterparts**
>
> **A1:** We conduct an ablative experiment on Human3.6M 2D pose estimation using 2D lifting. To build the 2D lifting baseline, we first remove the depth channel of 3D skeleton labels, and then change the output dimension to fit that. Other parts of the framework and hyper parameters are all kept unchanged. We use two-stage pre-training "IN+CC". The result is 87.6% JDR, 0.4% JDR below 3D lifting, showing that 2D lifting is inferior to 3D lifting on Human3.6M 2D pose estimation. We also evaluate it on MPII 2D pose estimation, and it achieves 88.7% PCKh\@0.5, which is 0.2% PCKh\@0.5 below that using 3D lifting.
>
> **Q2: An ablation study on the effectiveness of the KCS layer**
>
> **A2:** To conduct this ablative experiment, we remove the KCS branch in the Discriminator. The result on Human3.6M 2D pose estimation is 87.3% JDR, 0.7% JDR below full branch. The result is in accordance with our intention that KCS can help "better detect properties of human skeletons such as kinematic chains, bone lengths and joint angle limits" (in Sec 3.3).
>
> **Q3: Reproducibility and codes**
>
> **A3:** We appreciate your approval for our work's reproducibility. Training codes and pre-trained models will be released in GitHub for future work. Moreover, what we expect more is that our LiftedCL can inspire future works when learning 3D-aware representations for other tasks. For instance, learning 3D-aware representations for object detection. We have shown the feasibility to learn via lifting and adversarial learning in human centric tasks, however, how to do it for other tasks is not very clear right now and it could be a valuable research subject.

---

### Decision · Program_Chairs · 2023-01-20

**Decision:**

Accept: poster

**Justification For Why Not Higher Score:**

The key ideas of this work seem to be coming from other works and might be incremental, e.g., the usage of contrastive loss and 3D structure representation of humans.


**Justification For Why Not Lower Score:**

This paper shows strong empirical experimental results, which could be helpful to further development in the community.


**Metareview: Summary, Strengths And Weaknesses:**

This paper was reviewed by three experts in the field. Based on the reviewers' feedback, the decision is to recommend the paper for acceptance to ICLR 2023. All the reviewers acknowledged the good execution of leveraging invariance and equivariance for learning 3D-aware human-centric representations. The reviewers did raise some valuable concerns that should be addressed in the final camera-ready version of the paper, e.g., adding stronger baselines in which all the 3D elements are replaced by 2D counterparts. The authors are encouraged to make the necessary changes to the best of their ability. We congratulate the authors on the acceptance of their paper!


**Note From Pc:**

if the above contains the word "oral" or "spotlight" please see: "oral" presentation means -> notable-top-5% and "spotlight" means -> notable-top-25%. As stated in our emails, we are disassociating presentation type from AC recommendations

**Summary Of Ac-Reviewer Meeting:**

N/A